# Laser-Activated Second Harmonic Generation in Flexible Membrane with Si Nanowires

**DOI:** 10.3390/nano13091563

**Published:** 2023-05-06

**Authors:** Viktoriia Mastalieva, Vladimir Neplokh, Arseniy Aybush, Vladimir Fedorov, Anastasiya Yakubova, Olga Koval, Alexander Gudovskikh, Sergey Makarov, Ivan Mukhin

**Affiliations:** 1Center of Nanotechnology, Alferov University, Khlopina 8/3, 194021 St. Petersburg, Russiaimukhin@yandex.ru (I.M.); 2Higher School of Engineering Physics, Peter the Great St. Petersburg Polytechnic University, Polytechnicheskaya 29, 195251 St. Petersburg, Russia; 3N.N. Semenov Federal Research Center for Chemical Physics, Russian Academy of Sciences, Kosygin Street 4, 119991 Moscow, Russia; 4Moscow Institute of Physics and Technology, Center for Photonics and 2D Materials, Moscow Institute of Physics and Technology, 9 Institutskiy Lane, 141701 Dolgoprudny, Russia; o.yu.koval@gmail.com; 5School Department of Physics and Engineering, ITMO University, Lomonosova 9, 197101 St. Petersburg, Russia; 6Qingdao Innovation and Development Center, Harbin Engineering University, Qingdao 266000, China

**Keywords:** nonlinear nanophotonics, second harmonics, infrared imaging, flexible devices, nanowires

## Abstract

Nonlinear silicon photonics has a high compatibility with CMOS technology and therefore is particularly attractive for various purposes and applications. Second harmonic generation (SHG) in silicon nanowires (NWs) is widely studied for its high sensitivity to structural changes, low-cost fabrication, and efficient tunability of photonic properties. In this study, we report a fabrication and SHG study of Si nanowire/siloxane flexible membranes. The proposed highly transparent flexible membranes revealed a strong nonlinear response, which was enhanced via activation by an infrared laser beam. The vertical arrays of several nanometer-thin Si NWs effectively generate the SH signal after being exposed to femtosecond infrared laser irradiation in the spectral range of 800–1020 nm. The stable enhancement of SHG induced by laser exposure can be attributed to the functional modifications of the Si NW surface, which can be used for the development of efficient nonlinear platforms based on silicon. This study delivers a valuable contribution to the advancement of optical devices based on silicon and presents novel design and fabrication methods for infrared converters.

## 1. Introduction

Highly developed and widely used semiconductors with crystalline centrosymmetry, such as silicon [1,2] and germanium, have unfortunately zero second-order bulk nonlinear susceptibility $\chi^{\left(2\right)}$, hindering their developments in nonlinear photonics [3]. Second and third harmonic generation (SHG and THG) [4,5,6] is typically studied in thick materials with high nonlinearities [7,8,9,10] and the under phase-matching conditions required to achieve a high harmonic generation efficiency. However, the practical need for transducers of infrared (IR) radiation into the visible range [11] and photonic platforms integrated with CMOS requires developed nanoscale geometry and new functionalities such as flexibility and semitransparency. Most existing flexible nonlinear IR-to-Vis converter structures (used for laser setup alignments, etc.) are known to have a number of drawbacks, including the following: a lack of material transparency, limited range, non-environmental materials, or the need for light activation. Alternative device designs featuring semitransparent GaP nanowire/siloxane membranes [12] have proven highly promising; therefore, in this study, we explore similar structures but with more convenient Si material [13,14].

Arrays of silicon NWs have THG [15] in the NW volume, while the NW surface provides a remarkable SHG signal [16]. The SHG can be associated with the violation of crystal symmetry on the silicon NW sidewall surface, and the high SHG efficiency in Si NWs is attributed to a high surface-to-volume ratio in the NW geometry. Thus, Si NW arrays with a large surface area embedded in optically transparent silicone are promising objects for studying nonlinear optical phenomena. Due to the developed surface of NWs, such membranes allow for the study of SHG and can be used for integration with silicon electronics, including the development of optical information transmission technologies [17,18].

In this study, we fabricate Si NWs/siloxane membranes and present the study of SHG via IR femtosecond laser irradiation, for the first time, investigated in the proposed structure design. We observe a stable laser-induced enhancement of SHG, which is attributed to functional modifications of the Si NW surface. The proposed approach has the potential to aid in the development of efficient nonlinear platforms based on silicon nanostructures [19].

## 2. Experimental Section

### 2.1. Si NWs Fabrication

The SiNW array was formed by dry cryogenic etching using inductively coupled plasma (ICP) Oxford PlasmaLab ICP 380 setup [20,21,22]. The 4-inch (100) silicon wafers were preliminarily cleaned from organic contaminants and loaded to the etching chamber via load lock. The gas mixture of SF_6_ and O_2_ with a pressure of 5 mTorr was used for dry etching. The process was performed at a temperature of −150 °C to achieve over passivation [23]. The ICP power was set equal to 1000 W while additional RF power of 40 W was applied to the substrate holder.

### 2.2. Membrane Fabrication

The arrays of SiNWs were then encapsulated into polydimethylsiloxane (PDMS) Sylgard 184 Dow Corning via the G-coating method, allowing for the PDMS material to be penetrated between NWs and their bases [24,25,26,27]. After PDMS deposition, the samples were baked on a hot plate at 120 °C for 8 h. The SiNWs array was encapsulated into a PDMS layer via the G-coating method. G-coating is performed using a swing bucket centrifuge instead of a standard spin-coater. For sample processing, we used a high-speed Eppendorf 5804 centrifuge. This method allows for G-coating the PDMS, where the thinning force is normal for the substrate plane (along the NW axis) in contrast to the standard spinning, in which the thinning force lies in the lateral plane (perpendicular to the NW axis).

After the coating, the samples were placed on a hot plate at 120 °C to cure PDMS. This way, the NWs array remains vertically oriented after the PDMS is cured, so the NWs can no longer change their orientation and position. To improve the mechanical stability of the Si NW/PDMS samples and facilitate further manipulations, the membranes were treated in an oxygen plasma and stuck to a 200–400 μm thick PDMS cap film. Then, the Si NW/PDMS membranes were released from the growth substrate by a microtome razor blade (see Figure 1).

### 2.3. XRD Experiment

The crystal structure of the Si NW/PDMS membranes was investigated using the X-ray diffraction reciprocal space mapping (XRD-RSM) technique on a laboratory X-ray diffractometer. Diffraction images were obtained using a Bruker Kappa Apex II diffractometer equipped with a microfocus Incoatec IμS Cu Kα_1,2_ X-ray source (λ ~ 1.542 Å, beam size of 400 μm at the sample position) and a 2D charge-coupled detector. Si NW/PDMS membrane samples were mounted with the surface normally oriented along the diffractometer φ-axis. 3D reciprocal space maps were acquired by performing a single 360° sample azimuthal rotation with an angular step of 0.5° (φ-scan). φ-scans were performed at a fixed glancing angle of 30° or 9°, depending on the sample. Exposure time ranged from 10 to 30 s. Reciprocal space intensity distribution was reconstructed using RecSpaceQt software [28]. The silicon lattice crystallographic information file (9011998.cif) was downloaded from the Crystallography Open Database (COD). 

### 2.4. Second Harmonic Measurements

Nonlinear optical studies of Si NW/PDMS membranes were performed using a laser-scanning microscope (LSM-980, Zeiss, Germany). The external AOM port of the LSM was used to supply femtosecond laser pulses (Discovery-NX, Coherent, USA, California, Santa Clara, CA 95054) with the following characteristics: (1) repetition rate of 80 MHz; (2) duration of ~150 fs (spectral full width at half maximum (FWHM) < 10 nm); (3) linear polarization. Tunable wavelength range of 800–1020 nm for the central wavelength of the pulses was restricted by LSM optics. The final power of laser radiation supplied on the sample was measured using an Si detector (S120C, PM100USB, Thorlabs, Munich, Germany) and calibrated for AOM levels of LSM. The maximum power of radiation can be up to ~25 mW (0.3 nJ per single pulse), but for this study, it was in the range of 0.1–10 mW. Air microscopy objectives with numerical apertures of 0.3 and 0.8 (magnifications of 10× and 20×, respectively) were used in optical study. 16-bit images in LSM measurements were obtained using a sensitive GaAsP PMT detector, which allowed for the use of the galvo mirror system of the LSM at an average speed of tens of mm per second (~20 µs for optical lateral resolution of ~500 nm for 20× objective). The integral signal of the second harmonic was collected in the range of ±10 nm with respect to the central frequency of the harmonic for each pixel of the LSM image. We employed the lambda LSM scans to study possible nonlinear responses for different parts of the visible wavelength range, with a resolution of 3 nm.

## 3. Results and Discussion

The SiNW array morphology was studied via SEM. The SEM images presented in Figure 1 demonstrate a dense (>10 NW/µm^2^) array of narrow (about 30–80 nm lateral size) SiNWs with an ahomogeneous length of 1.1 µm.

The SiNW/PDMS membrane samples were fabricated from multiple pieces of SiNW array on an Si substrate of different dimensions. The smaller samples were used for the peeling tests. We used pieces of about 10 mm^2^, which provided us with the idea of a proper peeling technique suitable for the given NW morphology and density. Then, the efficient techniques were tested with larger samples of SiNW array, i.e., pieces of about 0.5 cm^2^ area. It was found out that the peeling process is very sensitive to the blade homogeneous sharpness and the Si wafer roughness (originated from the vertically inhomogeneous NW etching). This sensitivity leads to a partial transfer failure, when the SiNW/PDMS membrane partly remains on the Si substrate at areas where the surface roughness led to the local disconnection of the peeling blade. Therefore, the peeling technique is required to be relatively insensitive to the sample surface roughness, and it was found out that it can be achieved by heavy and hard blades, such as stainless steel microtome blades weighing about 0.5–1 kg.

However, even the best developed peeling technique cannot provide a complete transfer of a 5–7 cm^2^ SiNW/PDMS samples. As shown in Figure 1d, even the best SiNW/PDMS membrane sample of 4 cm^2^ demonstrated only partial transfer, and the effective transfer area was about 60–70 per cent. Based on our experience, we consider the main reason for this partial transfer failure to be the high adhesion of the PDMS material to the plates of the razor blade. This adhesion leads the released membrane to stick to the peeling blade and roll instead of smoothly gliding. Teflon coating of the blade plates can improve the situation, and as a consequence, the peeling results. However, the peeling method still requires further development, especially for the large area (above 50 cm^2^) SiNW/PDMS membranes. The analysis of the XRD-RSMs obtained from the as-formed Si NW/PDMS membranes allowed us to study the crystal structure of Si NWs. Diffraction spots, presented in XRD-RSM, were identified and indexed as those formed from the Si lattice with a [001] direction oriented normally to the membrane surface. Two-dimensional cross-sections of RSM cuts were analyzed to determine the presence of any other crystalline phases. Two orthogonal RSM cross-sections (ΔQ = ±0.05 Å^−1^) were plotted to illustrate the Si lattice orientation in relation to the PDMS membrane plane. Figure 2 shows the 2D reciprocal space intensity distributions in a plane intersecting the reciprocal space origin with a plane normal along the [110]_Si_ zone axis and in the (hk3)_Si_ reciprocal space plane, respectively, obtained for the as-formed sample.

Thus, we conclude that the ICP etching retains the initial substrate crystalline structure, ensuring it is unchanged, so the Si NWs have a monocrystalline silicon structure with a 001 orientation along the NW axis. This orientation coincides with the orientation of the Si(001) substrate on the surface on which the SiNWs were etched.

It should be noted that mechanical peeling of the Si NW/PDMS membranes from the Si wafer with a razor blade disrupts the initial ordering of Si NWs in the array (the path of the microtome razor blade is shown by the black arrow in Figure 2b inset). On the other hand, thin Si NWs can bend during the ICP etching process. The misorientation of the Si NWs in PDMS membrane led to an elongation of the Bragg reflexes in the reciprocal space along the circular paths, which is clearly visible in the reciprocal space intensity distribution in (hk3) plane. Black arrows in Figure 2a,b indicate an elongation of the diffraction reflection due to the misorientation of Si NWs. Thus, the PDMS encapsulation and membrane release technique generally preserve NW verticality, which is important for further optical studies.

Given the possibility of the SHG signal from an array of SiNWs on a substrate before release, we have not provided data from SHG measurements because the SHG signal from NWs before release is always many orders below the SiNW/PDMS membranes, and the SHG signal and its interpretation is incomparable for SiNW arrays before and after release from the substrate. Indeed, the Si substrate is an excellent sink to the generated SH in the visible spectrum due to its high refractive index, leading to the light soaking inside the wafer for further absorption. The Si NWs and the substrate are one whole. It is known that the EM wave appearing in SiNWs goes through an optically dense medium, so it is likely to propagate inside the substrate of the same refractive index. On the contrary, When the Si NWs are encapsulated in PDMS and released, the SHG signal is effectively scattered and can be efficiently detected.

The SHG measurements were limited to a range of 800–1020 nm for the fundamental wavelength due to the setup limitations. We expected the Si NWs in the PDMS membrane to generate the SHG signal at a relatively high pump intensity, so the NWs are at risk of overheating and destruction due to their low heat capacity and the weak heat evacuation by PDMS membrane material [29]. However, the SHG signal was detected at a pump power only one order higher in comparison with the measured reference sample of GaP NW/PDMS membrane, which was characterized by a record high SHG efficiency [12]. First, the SHG regime was confirmed by the spectral measurements. The nonlinear response from the Si NWs followed the laser pump with a distinct narrow spectral line, indicating the SHG signal dominated over the multiphoton photoluminescence. The spectral dependence of SHG is presented in Figure 3.

The spectral dependency of the SHG signal in the range of (800–1020) nm for several bright spots in the measured SHG map are presented in Figure 4. The shape of curves is reproducible for all chosen spots, with the maximum being in the range of 910–940 nm and having slightly asymmetric shoulders.

The confocal SHG mapping setup also provided a 3D visualization of the SHG in the studied area, constructed from the stack of 2D maps measured with a consequent focal plane shift (Figure 5). This 3D map confirms an inhomogeneous signal from the SiNW array; however, the confocal optical system has a relatively low resolution in both lateral plane (about 300 nm) and focal axis (about 1–2 µm), which is above or comparable with SiNW dimensions. Therefore, the detailed information where the SHG originated from (i.e., the NW sidewalls or the volume), as well as the SHG homogeneity along the NW axis, cannot be extracted. We associate the observed inhomogeneity of SHG in the NW array with the inhomogeneous distribution of the Si NW diameter, leading to a decrease in the active material volume.

The 3D visualization also confirmed that the SHG signal is very sensitive to the pump laser focal plane. As shown in Figure 5, the laser focal plane should be exactly in the plane of SiNW, and at just a few hundred nm defocus, the signal completely disappears. We expected, indeed, that the pump should be focused to create the threshold power density required for the SHG; however, we considered a few µm lag, comparable with the SiNW length, is tolerable since the SHG originates from the SiNW material and it is homogeneous along a few µm-long NWs. This sharp dependency on the focal plane position can be attributed to the NW resonant properties. We speculate that the NW array exhibits waveguiding (light-trapping) properties, so the pump laser beam focused exactly in the plane of the SiNW tips is effectively pulled inside the NW array, while the defocused laser beam (e.g., when the laser focal plane is inside the NW array, i.e., below the plane of the SiNW tips) is effectively scattered by the SiNW array, so the pump power dispersed instead of focusing.

However, we did not perform a systematic study of the SHG dependency on the pump laser beam focal plane position. All the presented results were achieved at optimal condition, i.e., when the focal plane is adjusted to the maximum SHG signal. The resonant pump laser beam focus can be further increased for the SiNW array of a higher NW diameter (above 400–600 nm), when the incident light can tangle with the self-modes of the NW cylinder resonator. In this case, the direct aiming of the pump laser focal plane at the plane of SiNW top facets is critical for the resonance SHG conversion.

The SHG–pump power measurements were performed at the optimal wavelength of 920 nm. This revealed an unexpected phenomenon in the form of a dramatic increase in the SHG signal at a specific threshold pump power. Namely, the SHG signal abruptly increases when the pump power reaches the specific value, which is the same for all tested regions of the studied SiNW/PDMS membrane samples. After this ‘activation,’ the studied area continues to demonstrate the enhanced SHG signal even below the threshold pump power, i.e., when the SHG signal acquisition is repeated at the previously measured pump power value below the threshold. This phenomenon can be called the activation effect, and it can be clearly seen from the SHG map presented in Figure 5 that it is acquired at a pump power value that is below the threshold, while the area on the map marked by red rectangle corresponds to the sample region previously ‘activated’ at the threshold pump power value.

To study this phenomenon in detail, the SHG–pump power study was performed in a repetitive manner. First, the original sample area was mapped at the pump power rising from the minimal value (when the SHG signal can be detected) to the maximum value of the pump power range (defined in advance as the highest pump power before the structure degradation due to the overheating). Second, the same area was tested at the descending pump power at the same power range. Finally, the first run was repeated, i.e., the pump power was increased again from the minimal to the maximal value but at the already exposed area. The acquired data were processed to draw curves for selected bright spots corresponding to the most efficient individual Si NWs (Figure 6a–c) and the entire map area to determine the Si NW array mean SHG-to-pump value dependency (Figure 6d–f) on a double logarithmic scale.

As can be expected, the slope value for the initial measurements at the virgin area, when the activation effect occurs, is significantly different from the characteristic SHG value of 2. Indeed, it varies from 3 to 3.6 for the individual NWs, and is about 2.1–2.2 for the averaged signal. Meanwhile, the SHG-to-pump slope for the repeated measurements after the first run is very close to 2 for both the individual Si NWs and the array (Figure 6). This led us to the conclusion that the initial ‘activation’ measurements are characterized by some sort of transition processes in the Si NWs, so the SHG efficiency improves with increasing laser pump power. Once the maximum pump power is reached at the first run, this transition process is completed, and the exposed SiNWs generate SH at a normal efficiency.

We speculate that this transition process can be explained either by the structural (material) transformations in the Si NWs induced by the pump power (e.g., monocrystalline(mc)-Si transforms in polycrystalline-Si [30] or surface oxidation [31]), or the functional transformations (e.g., the trapping states on the NW surface capture the carriers excited by the laser beam leading to the build-in electrostatic field appearance), or both material and functional.

Taking into account the centrosymmetric structure of Si ($\chi_{Si}^{\left(2\right)}=0$), the SHG signal from Si NWs can be described as $I_{2\omega }∼{\left|\chi_{Si}^{\left(2\right)}+\chi_{Si}^{\left(3\right)}E_{dc}\right|}^{2}I_{\omega }^{2}∼{\left|\chi_{Si}^{\left(3\right)}E_{dc}\right|}^{2}I_{\omega }^{2}$ [32], where $E_{dc}$ is a build-in field at the surface of NW (electrical field induced second harmonic, EFISH [33]). At static, $E_{dc}∼\frac{\Delta \varphi }{d}$ is constant, where Δφ and d are the surface potential barrier height and width of the depleted area, respectively. Therefore, the slope value is close to 2.

Si NWs represent thin silicon material nanoscale crystals with a high surface-to-volume ratio. Therefore, the well-known Si surface modifications induced by the intense laser beam irradiation can dramatically affect the SHG process. The surface oxidation or other material process under femtosecond laser irradiation is expected, and it can modify the near-surface SiNW material at the depth of several tens of nanometers [16]. Thus, the centrosymmetric mc-Si can transform into another phase, having lower symmetry in comparison with the cubic c-Si and, therefore, the non-zero second-order susceptibility.

Considering the transition effect, the dependence of SHG signal changes according to the following expression $I_{2\omega }∼{\left|x_{nph}^{\left(2\right)}\right|}^{2}I_{\omega }^{2}V+{\left|\chi_{Si}^{\left(3\right)}E_{dc}\right|}^{2}I_{\omega }^{2}V_{0}$, where $\chi_{\mathrm{nph}}^{\left(2\right)}$ and V are the second order susceptibility (not equal to 0) and volume of new phase material, $V_{0}$ is the volume of Si NWs. Note that the volume of new phase can increase with power pump $I_{\omega }$, since the laser-induced material transformation can continue ($V∼I^{\alpha }$).

Moreover, due to the transition process, $E_{dc}$ can also depend on $I_{\omega }$, giving a super quadratic dependence of $I_{2\omega }$ on $I_{\omega }$. Under laser radiation, the oxidized silicon surface or other material surrounding the NW (e.g., Si, C, SiO_x_ or SiC formed under fs-laser irradiation) can trap light-excited carriers, changing the height of surface potential barrier in Si NW. At the same time, it is well known that the effect of laser-induced oxidation of NWs [34,35] leads to the enhancement of thickness in the NW oxide layer, and, as a sequence, changes the build-in field ($\Delta \varphi =\frac{Q}{C}$, $C∼\frac{1}{a}$, Δφ∼a, where Q, C, and a are the trapped charges, capacity, and thickness of the oxide layer, respectively). Thus, with the increase of $I_{\omega }$, the build-in field $E_{dc}$ can also nonlinearly increase ($E_{dc}∼I^{\beta }$). Accounting for these possible mechanisms, we can suggest the following power dependence $I_{2\omega }∼I_{\omega }^{2+\gamma }$ (where γ > 0 and is defined by the dominating nonlinear process), which aligns well with our experimental results.

The transition is finished after the initial SHG-to-pump measurements; thus, the exposed area generates a normal SH signal characterized by the typical slope value of 2.

## 4. Conclusions

We have demonstrated an enhanced generation of the visual second harmonic by the proposed Si NW/PDMS membranes. The SHG can be attributed to the Si NW near-surface mc-Si material, and it is facilitated by the intrinsic electric field by the EFISH mechanism. In the experiment, we observed the effect of fs-laser irradiation, leading to an abrupt and permanent increase in the SHG signal, which we associate with the restructuring of the surface material and EFISH due to the charging of the near-surface electronic trap states that is proven by SHG-to-pump slopes of around 3.5.

As a result, the proposed Si NW/PDMS membrane, due to the laser activation effect, can serve as infrared-to-visual range converters and would be in demand for the advanced optical schemes where efficient, semitransparent, permanent, flexible, and non-disturbing control of the laser beam is of high importance, as well as in the general field of CMOS-integrated Si-based photonic circuits.

Si NW/PDMS membranes are promising materials for studying the nonlinear optical effects at nanoscales, and could have a variety of uses in sensor applications due to the large surface-to-volume ratio of Si NWs. Si NWs can be used for nonlinear light manipulation in future integrated optoelectronics (CMOS-integrated Si-based photonic circuits).

## Figures and Tables

**Figure 1 nanomaterials-13-01563-f001:**
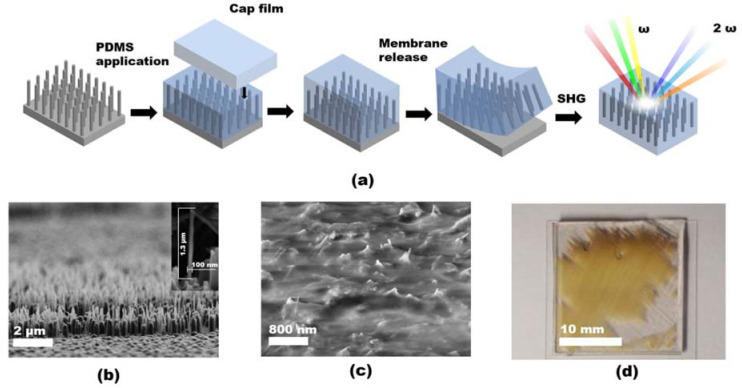
(**a**) Processing schematics of SiNW/PDMS membrane. (**b**,**c**) SEM images of as-formed SiNW array and encapsulated into PDMS. (**d**) Photo of the SiNW/PDMS membrane after release from the Si substrate.

**Figure 2 nanomaterials-13-01563-f002:**
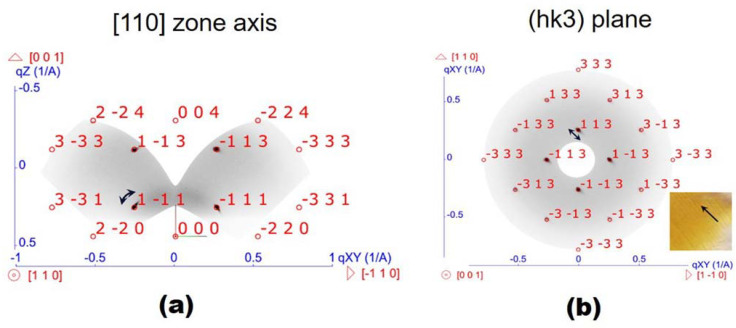
Orthogonal 2D cross-sections (ΔQ = ±0.05 Å^−1^) of the reciprocal space intensity distribution (in linear intensity scale) obtained from the as-formed SiNW/PDMS membranes, (**a**) [110]_Si_ zone axis and (**b**) (hk3)_Si_ plane. The lattice directions and model positions of the reciprocal lattice nodes of silicon are shown by the red arrows and indexed circles, respectively. A microphotograph of the studied sample is shown in the inset.

**Figure 3 nanomaterials-13-01563-f003:**
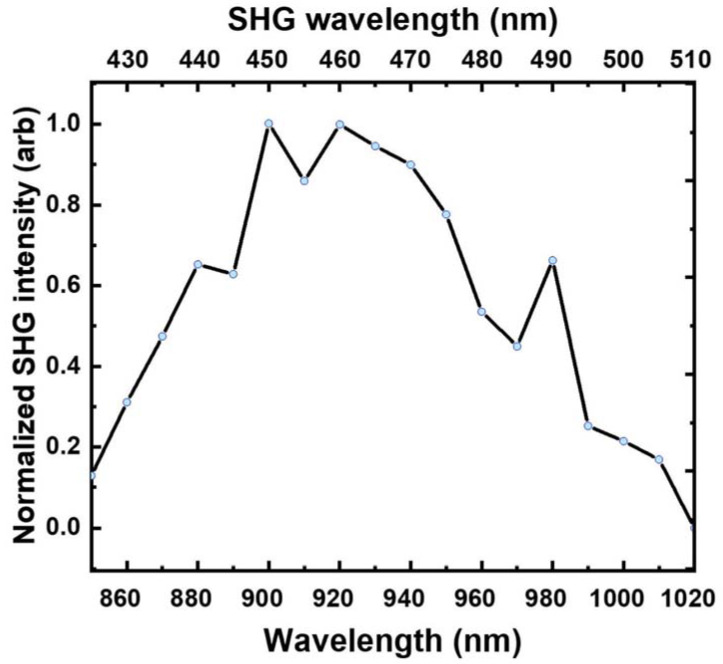
SiNW SHG intensity dependence on wavelength.

**Figure 4 nanomaterials-13-01563-f004:**
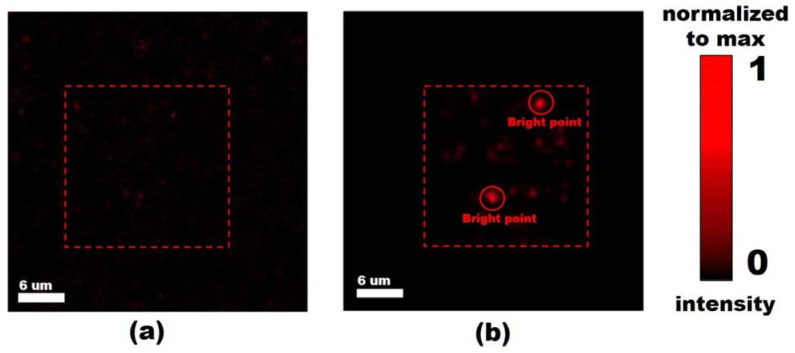
SHG maps for the SiNW/PDMS membrane area (**a**) before and (**b**) after laser activation. The labeled spots in the map (**b**) correspond to SiNWs effectively producing SH.

**Figure 5 nanomaterials-13-01563-f005:**
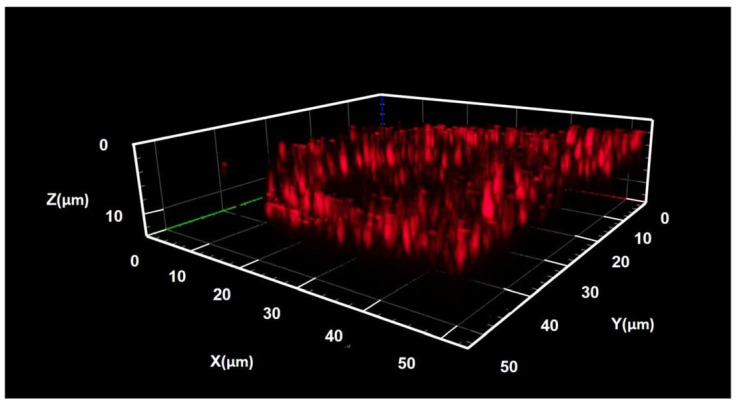
3D visualization of the experimental SHG signal from the SiNW/PDMS membrane. Distribution of the signal intensity over the studied membrane area (z-stack).

**Figure 6 nanomaterials-13-01563-f006:**
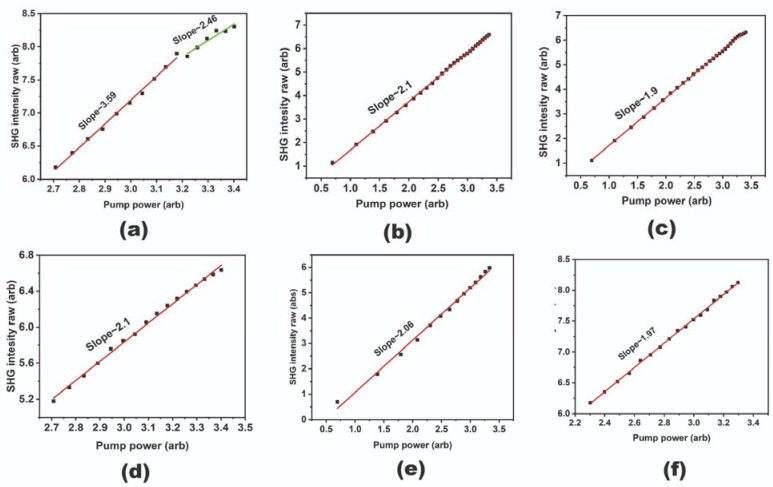
SHG-to-pump curves and the corresponding slope values. (**a**–**c**) Figures correspond to a representative bright spot in the SHG map, measured one after another. (**d**–**f**) Correspond to the integral signal over the studied area in the map.

## Data Availability

Not applicable.

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
