# Peer review of "Laser-Activated Second Harmonic Generation in Flexible Membrane with Si Nanowires"

_nanomaterials, 2023, doi:10.3390/nano13091563_

Round 1
Reviewer 1 Report
In the manuscript, the authors have demonstrated an enhanced generation of the visual second harmonic by the 256 proposed SiNW/PDMS membranes. Moreover, the SHG can be attributed to the SiNW near-surface 257mc-Si material, and it is facilitated by the intrinsic electric field by the EFISH mechanism.The manuscript presents some interest for researchers in the area of nonlinear silicon photonics. In my opinion, this manuscript can be accepted and published with some revision. However, before that, some questions should be corrected.
1. What does the obtained result reveals?
2. What is your novelty compared to previous studies? Are the generated SHG essentially different from those of previously reported?
3. What is the practical application of the laser-activated second harmonic generation in flexible membrane with Si nanowires?
4. The quality of the English should be revised carefully.
The quality of the English should be revised carefully.
Author Response
Dear reviewer, the answers to the questions are given in a separate document. Please see the attachment.

Reviewer 2 Report
The manuscript deals with a subject that certainly has some relevance and that can be of interest to a fraction of readers.
It can be published in this journal but after some revision because it is a bit difficult to read since the ideas are not well explained for most of the readers and some figures, particularly Fig 3 are extremely difficult to understand. What are is the difference between right and left scales? for example, both labelled as SHG intensity.
Also typing needs to be more careful to keep the excellent stile of this journal, as an example what means 800..1200, maybe you mean 800 ... 1200?
Author Response

(The authors gave the same response as above.)

Author Response

(The authors gave the same response as above.)
